# Disorder enabled band structure engineering of a topological insulator surface

Yishuai Xu[1], Janet Chiu[1], Lin Miao[1,2], Haowei He[1], Zhanybek Alpichshev[3,4], A. Kapitulnik[4], Rudro R. Biswas[5] & L. Andrew Wray[1,6]

Three-dimensional topological insulators are bulk insulators with $Z_2$ topological electronic order that gives rise to conducting light-like surface states. These surface electrons are exceptionally resistant to localization by non-magnetic disorder, and have been adopted as the basis for a wide range of proposals to achieve new quasiparticle species and device functionality. Recent studies have yielded a surprise by showing that in spite of resisting localization, topological insulator surface electrons can be reshaped by defects into distinctive resonance states. Here we use numerical simulations and scanning tunnelling microscopy data to show that these resonance states have significance well beyond the localized regime usually associated with impurity bands. At native densities in the model $Bi_2X_3$ (X = Bi, Te) compounds, defect resonance states are predicted to generate a new quantum basis for an emergent electron gas that supports diffusive electrical transport.

[1] Department of Physics, New York University, New York, New York 10003, USA. [2] Advanced Light Source, Lawrence Berkeley National Laboratory, Berkeley, California 94720, USA. [3] Massachusetts Institute of Technology, Department of Physics, Cambridge, Massachusetts 02139, USA. [4] Department of Physics, Stanford University, Stanford, California 94305, USA. [5] Department of Physics and Astronomy, Purdue University, West Lafayette, Indiana 47907, USA. [6] NYU-ECNU Institute of Physics at NYU Shanghai, 3663 Zhongshan Road North, Shanghai 200062, China. Correspondence and requests for materials should be addressed to R.R.B. (email: rrbiswas@purdue.edu) or to L.A.W. (email: lawray@nyu.edu).

Three-dimensional topological insulators (TI) with the chemical formula $M_2X_3$ ($M = Bi$, Sb; $X = Se$, Te) are characterized by a single-Dirac-cone surface state[1-3], and have been widely investigated as model materials for TI surface physics. Shortly after the discovery of the first TI's, it was established that standard effects of defect disorder such as backscattering and localization are greatly suppressed in TI surface states[4-11]. Altogether with the intuition that topological properties of band structure should be inherently impervious to weak perturbations, this led to a pervasive characterization that topological interfaces are essentially blind to the presence of non-magnetic defects. It is only with the recent advent of more detailed non-perturbative numerical investigations and targeted scanning tunnelling microscopy (STM) experiments[12-25] that elements of this picture have been overturned. It is now known that, rather than being blind to defects, topological Dirac cone electrons actually bind loosely to atomic point defects in energy levels that fall very close to the Dirac point (see states labelled 'resonance' in Fig. 1b), and can radically redefine electronic structure relevant to the Dirac transport regime.

In isotropic Dirac models of TI surface states, strongly perturbing non-magnetic defects are theoretically expected to create circular s-wave resonance states, such as those plotted in Fig. 1a. Experimentally, these states have been observed in STM measurements of density of states (DOS) immediately above the Dirac point of topological insulator $Bi_2Se_3$ (see Fig. 1a (left) inset patch). Three bright spots in the STM image represent coherent interference with a slight $\lesssim 1\%$ admixture of f-wave symmetry (Supplementary Note 2) because of anisotropy not considered in the accompanying simulation. The combined impact of multiple randomly distributed defects on the electronic structure has not yet been evaluated, and is the subject of the present work.

Theoretical treatments of lattice defects are usually carried out in the single impurity approximation, since multiple impurities do not lead to emergent coherent bandstructures. Large modifications to the band structure of non-TI materials are only achieved through heterostructure growth or fractional alloying. The 'impurity bands' associated with electrons on minority impurity sites are generally disregarded as being non-ideal for electronic transport. These bands exhibit miniscule dispersions, and strong defect potentials or reduced dimensionality cause the electronic states to be Anderson localized[26,27] over relevant device lengthscales. A typical example is the well-studied manganese impurity band in $Ga_{1-x}Mn_xAs$, whose dispersion across the Brillouin zone is far smaller than its intrinsic width along the energy axis, meaning that the electrons are effectively localized to isolated lattice sites[28,29].

In the following, in contrast, we find that uncompensated lattice defects at low densities ($\rho < 0.1\%$) in non-amorphous TIs provide a mechanism for band structure engineering that is not shared in a conventional semiconducting system. Theoretically, it is expected that Anderson localization will be forbidden on the TI surface with non-magnetic impurities[30-34] and backscattering is suppressed relative to forward scattering[4,5,9], leading to greater itinerancy for impurity-derived band structure. Moreover, the electronic density of states falls to zero at a surface Dirac point, making it possible for defect resonance states to dominate the absolute density of states. In the following discussions we will show that these conditions enable new defect-derived electronic dispersions, providing viable channels for diffusive electrical transport. We also find that these emergent electronic band structures can superficially appear to be gapped, and resemble Dirac point anomalies that have been seen by angle resolved photoemission spectroscopy (ARPES) in

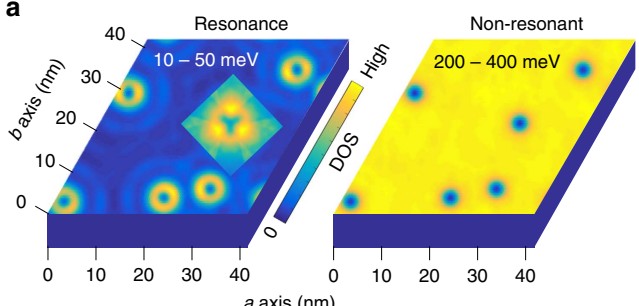

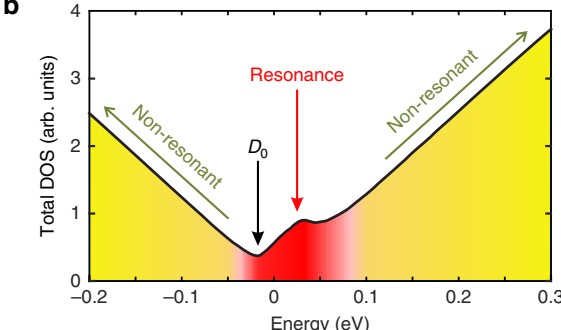

**Figure 1 | Defects on a TI surface.** (**a**) A simulation of density of states (DOS) in different energy regions at a TI surface with $100 \times 100$ lattice sites and six randomly placed point defects (density $\rho = 0.06\%$). One of the simulated defect resonance states has been overlaid with a three-fold symmetrized STM density of states image obtained 70 meV above the Dirac point of $Bi_2Se_3$. (**b**) The Dirac point energy ($D_0$) and resonance state peak are labelled on the energy-resolved DOS distribution for a large $350 \times 350$ site surface with $\rho = 0.06\%$ defect density. Red shading indicates states in which electrons adhere closely to defects.

TI compounds that have no known time reversal symmetry broken ground state[35-38].

## Results

**Defect resonance states of Dirac electrons.** We will focus on defect resonance states in the popular $Bi_2Se_{3-x}Te_x$ family of single-Dirac-cone TIs[1-3], so as to make quantitative comparisons to STM measurements. However, our results are applicable to other TI surface states with isolated Dirac cones. In this material family, structural point defects tend to occur with a density of $\rho \lesssim 0.1\%$ per formula unit, and have been found to overlap with the surface state in the outermost quintuple layer of the lattice[15,39] (see Supplementary Note 1). Intensive characterization of $Bi_2Se_3$ has found that the density and type-range of defects can be tuned over a broad range by adjusting sample growth conditions[39]. For Se-poor and stoichiometric synthesis, defects near a cleaved surface are predominantly selenium vacancies positioned in the centre of the outermost quintuple layer of the lattice. Interstitial atoms and replacement defects can be common at the surface of Se-rich or quench-cooled samples. The chemical potential can be shifted as a separate parameter via methods that include bulk doping, surface dosing, and electrostatic gating[40-43].

Defect resonance states have been observed by STM less than 200 meV above the surface Dirac point, surrounding point defects[14,15] and at step edges between plateaus differing in height by one quintuple layer[18,19]. Characteristic local density of states (LDOS) distributions measured by STM at the defect

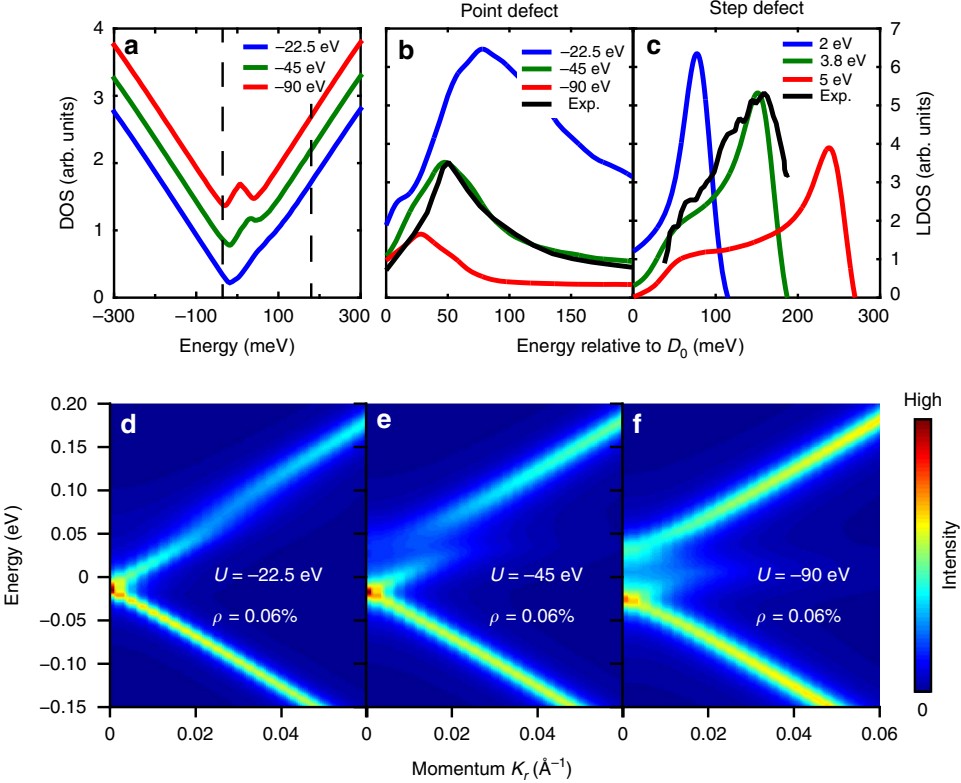

**Figure 2 | The non-magnetic defect potential.** (**a**) The DOS distribution for point defects with $\rho = 0.06\%$ density at different defect potentials. (**b**,**c**) The local DOS distribution of point ($\rho = 0.06\%$) and step defect resonance states at different '$U$' defect potentials. Black experimental curves are reproduced from STM investigations in refs 15,18. (**d**–**f**) The ARPES spectral function of a large TI surface with randomly distributed point defects is modelled as a function of defect potential.

lattice sites are plotted as black curves in Fig. 2b,c. Theoretical analyses of isolated local impurities have shown that these resonance states arise naturally when strong scalar potentials are introduced to a surface occupied by spin-helical TI Dirac cone electrons[12,13,15–17]. For an isolated defect described as a point potential with amplitude $U$, the resonance state is split from the Dirac point by an energy roughly proportional to $-U^{-1}$.

In the following, we numerically approach the problem of multiple impurity scattering on the surface states of TIs with a single Dirac cone, which is otherwise theoretically intractable. We use a basis of spinor plane waves[44] on a discrete hexagonal real-space lattice, with lattice parameter $a = 4.2$ Å corresponding to $Bi_2Se_{3-x}Te_x$ surface atomic spacings. We have used the impurity-free spin-helical Hamiltonian, $H_k = v_0(\mathbf{k} \times \sigma)$, with Dirac velocity $v_0 = 3.0$ eV·Å chosen to be intermediate between $Bi_2Se_3$ and $Bi_2Te_3$.

When a large surface with a typical $\rho \sim 0.06\%$ defect density is modelled in this way, we find that $\sim 2$ electronic states per defect are added near the Dirac point (see Supplementary Note 1; Supplementary Fig. 4), resulting in a new DOS peak that becomes increasingly visible as the defect potential grows (Fig. 2a). Isolating the LDOS at defect cores on $Bi_2Se_3$ reveals that a defect potential of $U = -45$ eV accurately reproduces the resonance state energy and LDOS line shape obtained in STM studies, which is plotted in Fig. 2b. The very different experimental line shape of a step edge defect can likewise be reproduced by considering the tunnelling barrier generated with a repulsive potential of $U = 3.8$ eV applied to each atom along the step edge path (Fig. 2c). These comparisons are evidence that our numerical approach is able to quantitatively capture the physics observed in experiments.

**New delocalized electronic structure.** The resonance states associated with this defect distribution reveal a novel structure in the momentum-resolved electron annihilation (ARPES) spectral function, modelled in Fig. 2d–f. The resonance itself shows up as a discontinuity in the ARPES spectral function. As the defect potential strength increases, the surface Dirac point (termed $D_0$), as well as the discontinuity associated with the resonance state LDOS maximum ($E_R$) move down in energy (Fig. 3h). When the defect potential crosses a critical value of $U \lesssim \sim 45$ eV, a new local maximum appears in the spectrum at $K = 0$ ($\bar{\Gamma}$ point), roughly 50 meV above the $D_0$ Dirac point, and will henceforth be referred to as '$D_1$'. As the defect resonance state converges on $E = 0$ eV with increasing potential strength, the region between $D_0$ and $D_1$ begins to qualitatively resemble a band gap, as has been seen at the Dirac point of TI crystals with non-magnetic defects in refs 35,36. Unlike a true band gap, which is theoretically disallowed[6], the momentum-integrated total DOS distribution contains a local maximum within the seemingly gapped region (Fig. 2a).

A remarkable evolution of the electronic structure also occurs when the defect density is increased (see Fig. 3a–c series). When the defect density is tripled to $\rho = 0.18\%$, the faint local maximum $D_1$ becomes the dispersion minimum of an upper band that is clearly disconnected from the $D_0$ Dirac cone. The $D_0$ upper Dirac cone maintains its dispersion for small momenta $K \lesssim 0.02$ Å, out to more than 50 meV from the Dirac point, and becomes dispersionless at larger momenta, as would be expected for a more typical impurity band. For all of these impurity densities, the upper $D_0$ Dirac cone dispersion is significantly larger than the energy axis intrinsic width, which is $\delta E \sim 0.02$ and 0.04 eV at half maximum intensity in Fig. 3b,c, respectively.

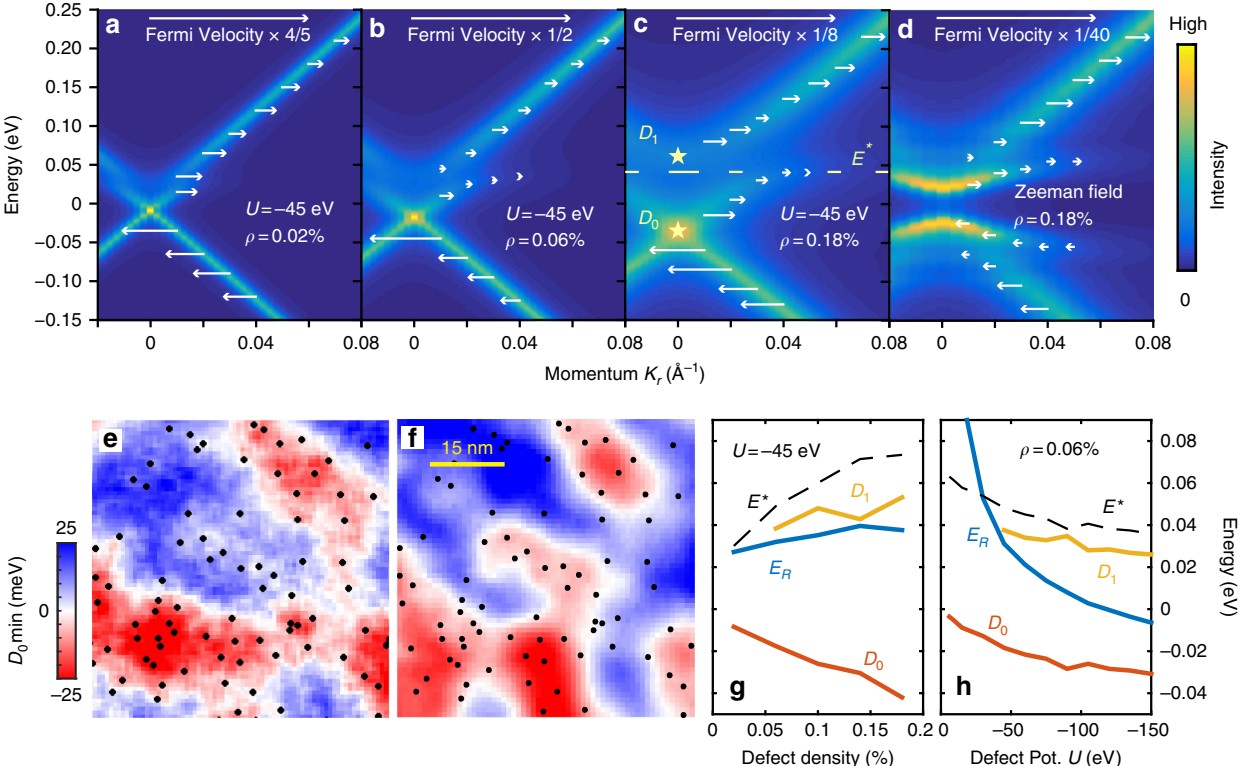

**Figure 3 | Emergent band structure. (a–c)** The momentum resolved spectral function of a large TI surface with randomly distributed point defects is shown as a function of defect density '$\rho$'. Arrows indicate radial axis twist velocity $v_\theta(E, \mathbf{K})$. The critical energy $E^\star$, resonance energy $E_R$ and $K = 0$ local maxima ($D_0$ and $D_1$) are labelled on panel **c**. **(d)** The non-magnetic twist velocity simulation in panel **c** is repeated for equivalently strong magnetic defects ($J = 45$ eV). **(e,f)** The spatial distribution of the LDOS minimum associated with the Dirac point is shown **(e)** from STM measurements on Se-poor $Bi_2Se_{3-\delta}$ and **(f)** from a simulation for $U = -45$, with black dots representing the locations of electron donating Se vacancy defects. Both distributions can be well represented in the same $\pm 25$ meV range. **(g,h)** The Dirac points, resonance state LDOS maximum, and critical energy $E^\star$ are plotted as a function of defect density and defect potential. The image in panel **(e)** is adapted with permission from ref. 39, copyrighted by the American Physical Society.

Higher defect densities are also associated with further lowering of the $D_0$ Dirac point energy. An STM map obtained on a Se vacancy-rich $Bi_2Se_{3-\delta}$ surface shows that spatial fluctuations in the energy of the $D_0$ LDOS minimum are highly correlated with defect density (Fig. 3e). Simulating the observed Se vacancies with a $U = -45$ defect potential reproduces the qualitative spatial distribution of the $D_0$ LDOS minimum (Fig. 3f). These modelling results can be plotted in the same $\pm 25$ meV range as the STM data, showing that the energy scale of fluctuations is accurately captured, even though fine details may be inconsistent because of missing information about defects outside of the STM map, or deeper in the crystal. We note that the spatial distribution of defects observed by STM is very uniform and uncorrelated, and is not readily distinguished from the random defect configurations used in our simulations (see Poisson distribution overlay in Supplementary Fig. 1).

The emergence of new spectral features at high defect density is closely related to the fraction of states near the Dirac point that can be attributed as defect resonance states. In the absence of defects, the total number of states within an energy $E$ of the Dirac point is $N_E = \frac{\sqrt{3}a^2 N E^2}{4\pi v_0^2}$, where $N$ is the total number of 2D surface lattice sites. Setting $N_E$ equal to the approximate number of resonance states ($N_E = N_R \equiv 2\rho N$) gives a new energy scale, $E^\star = \sqrt{\frac{8\pi v_0^2 \rho}{\sqrt{3}a^2}}$, which we find to have a novel physical significance. As can be see in Fig. 3g,h, we find that a local maximum associated with the emergent $D_1$ feature becomes

visible when the resonance state energy $E_R$ becomes smaller than $E^\star$. These simulations also reveal that the resonance energy may be tuned not only by changing the impurity strength, as has been theoretically predicted before, but also by increasing the impurity concentration, a fact that is technologically significant because the impurity strength is usually not tunable. With the experimentally fitted defect potential $U = -45$ eV, the resonance energy crosses $E^\star$ when the defect density becomes larger than $\rho = 0.02\%$, resulting in the easily visible emergent features in Fig. 3b,c.

To evaluate the ability of the emergent coherent impurity band-like features to carry mobile charge, we have calculated the twist velocity, $v_\theta$, for configurations with $300 \times 300$ sites ($300$ sites = 126 nm per side). These are shown in Fig. 3a–c, as vectors overlaid at local maxima of the modelled ARPES spectrum. The twist velocity of a quantum state is proportional to the magnitude of a persistent DC current carried by that state in the presence of an electric field (see Methods), and in the scattering-free limit is equivalent to the group velocity of the electronic band. It is calculated as the gradient of the state energies with respect to a phase twist around the system boundary.

We find that the twist velocities of band features at energies $E > 0$ eV have similar amplitudes, and are approximately proportional to the apparent slope in momentum space, suggesting that all of these features are viable bands for charge transport. However, twist velocities decay towards zero as the upper $D_0$ Dirac cone disperses into the seemingly gapped region

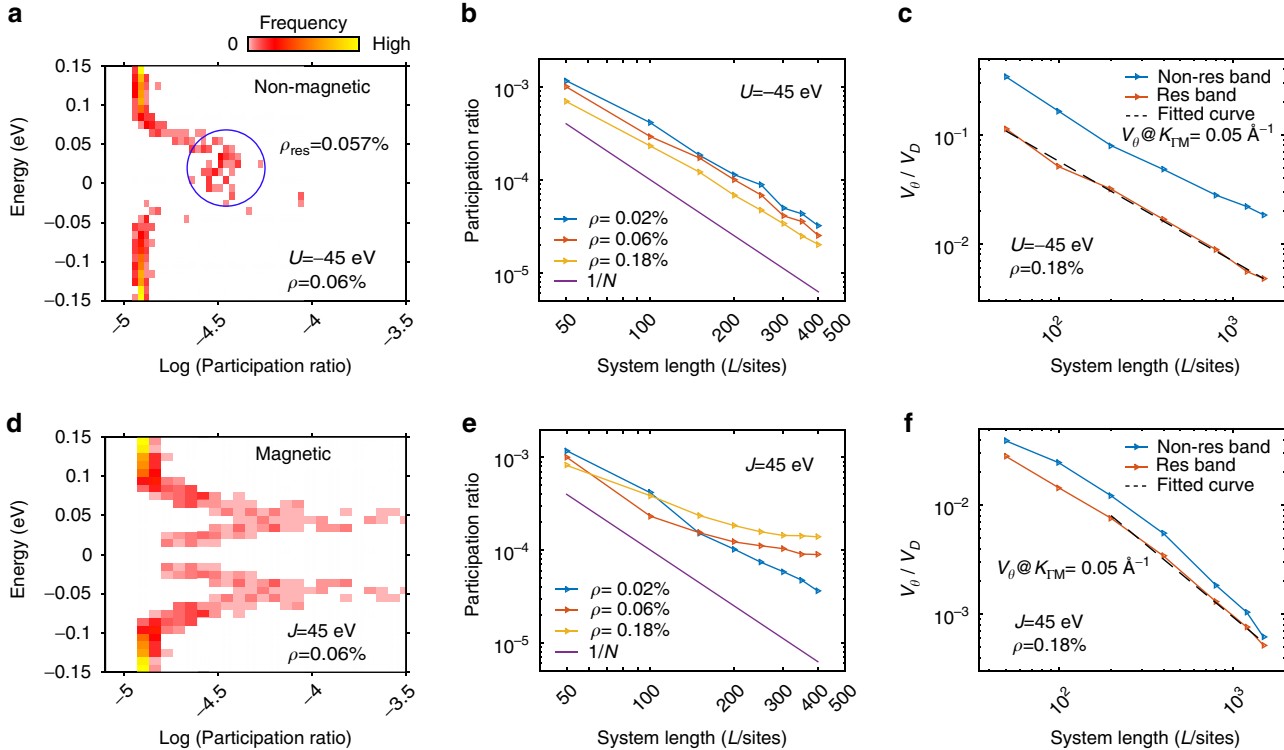

**Figure 4 | Real-space structure. (a)** A histogram of energy-resolved participation ratio for a $350 \times 350$ site simulated surface with $\rho = 0.06\%$ density scalar defects. The resonance region is circled, and labelled with the density of resonance states contained within the circle. **(b)** The dependence of resonance state participation ratios on the edge-length $L$ in an $L \times L$ simulation is plotted for different defect densities. **(c)** The dependence of twist velocity along a linear $L \times 100$ system at momentum $K = 0.05 \, \text{Å}^{-1}$ is shown for the resonance band and upper non-resonance band. The dashed fit curve is proportional to $L^{-0.91}$. Panels **d–f** show the same quantities for magnetic defects, using a fit curve proportional to $L^{-1.33}$ for **f**.

beneath the $D_1$ feature, particularly at momenta of $k \gtrsim 0.05 \, \text{Å}$, and this may greatly limit carrier mobility for certain values of the chemical potential (see evaluation of twist and band velocities in Supplementary Note 1; Supplementary Figs 2 and 3). Velocities are relatively large in the lower $D_0$ Dirac cone, where states are energetically distant from the resonance states, and maximal velocities are found immediately beneath $D_0$, where elastic scattering is minimal because of the vanishing density of states near a 2D Dirac point.

For comparison, a similar simulation with magnetic defects instead of a $U = -45 \, \text{eV}$ scalar field is shown in Fig. 3d, representing a case in which the topological protection against Anderson localization is expected to fail[45]. We have approximated each magnetic defect by a localized $J = 45 \, \text{eV}$ exchange field perfectly aligned with the $+z$-axis, perpendicular to the surface (see Methods). In this magnetic scenario, even though disorder of the magnetic moment vectors has been neglected, twist velocities are still reduced by an additional order of magnitude near the Dirac point because of Anderson localization.

We have also investigated the transport characteristics of the potential disorder-induced band structure by an alternate metric, the participation ratios of the quantum eigenstates. The participation ratio[46,47] of an electronic wavefunction is the sum of squares of the site-resolved probability densities (see Methods). For itinerant electronic states that can participate in electronic transport, it decays as the inverse of the total number of sites in the system ($N^{-1}$), for large system sizes. On the other hand, for localized states, the participation ratio converges on a non-zero value as system size increases beyond the localization length of the wavefunction. The participation ratios of the

energy eigenstates in the presence of scalar and magnetic disorder are compared in Fig. 4. We find that in the latter case, Anderson localized states around the resonances display large participation ratios that saturate with system size (Fig. 4e), while topological protection against localization ensures that the participation ratios for the comparable scalar potential disorder case vary inversely as the system size, for large system sizes (Fig. 4b). Calculating the twist velocities for larger system sizes shows a similar trend (Fig. 4c,f), in which scalar defect potentials allow diffusive transport ($v_\theta \gtrsim L^{-1}$), being topologically protected against Anderson localization, unlike conventional impurity bands, represented here by the same surface states in the presence of magnetic disorder of comparable strength.

## Discussion

Impurity-induced carrier control is the cornerstone of the semiconductor industry, and band structure engineering via disorder is the natural successor of such disorder-induced functionalization of electronic materials. In conventional 2D materials, Anderson localization prevents impurity bands from acting as viable carriers for diffusive charge transport. In this work, we have shown that this restriction is ameliorated in the surface states of topological insulators, in the presence of moderate potential disorder that does not break time reversal symmetry. Using numerical techniques verified by comparison to experimental observations, we have demonstrated that a new impurity band structure emerges in the TI surface states in the presence of scalar potential disorder, and that these states remain diffusive and are able to participate in transport processes. We have contrasted this novel behaviour, when topological

protection against localization is present, to the more conventional situation when magnetic disorder of comparable strength is present, when Anderson localization prevents diffusive transport in the impurity band.

Our results also shed light on some less understood DOS anomalies on TI surface states that have been reported in the literature. ARPES experiments have previously resolved gap-like features[35–37] and Dirac point elongation[38] in the spectrum of TI surface states, in the absence of any time-reversal breaking effects. The experiment in ref. 37 presents a particularly telling case. In this work, incident photon polarization was tuned to eliminate the dominant even-symmetry photoemission channel from $Bi_2Se_3$ along the high symmetry $\bar{\Gamma} - \bar{M}$ axis. Defect resonance states do not have spatial reflection symmetry because of their random distribution, and would not be intrinsically suppressed by this symmetry factor. This defect-sensitive measurement geometry revealed a gap-like dispersion, with the upper Dirac cone bands appearing to converge more than $E > 50$ meV above the Dirac point, as expected from our calculations for the $D_1$ feature.

We have shown that these observations may be attributed to the effects discussed above, arising from scalar potential disorder alone. The emergent band structure associated with defects is very narrowly separated from nearby bands in momentum, and will be difficult to cleanly resolve in ARPES experiments. The energy-axis widths of bands in our calculations are 2–3 times narrower than those actually observed in $Bi_2Se_3$, indicating that factors not considered here such as phonon scattering and surface inhomogeneity may play a significant role in the low energy physics, and may obscure the role of defect-derived states.

Nonetheless, the existence of defect resonance states immediately above the Dirac point of these compounds is well known from STM measurements, and the new physics discussed in this work are direct consequences of the interplay between these resonance states and other established elements of TI surface physics. The numerical evaluations of electron mobility and localization discussed above show that the emergent band structure identified in these simulations is viable as a basis for conductivity, and that band structure engineering is possible via non-magnetic defects in a TI system, driven by unique factors such as the lack of Anderson localization and the vanishing of DOS near a 2D Dirac point.

## Methods

**Energetics and state basis at the TI surface.** STM data and LDOS curves were obtained by the procedures described in refs 15,18. The surface state is modelled as a spin-helical 2D Dirac cone occupying a discrete hexagonal real-space lattice with repeating boundary conditions, and is essentially a discretized version of the model in refs 13,15. The kinetic Hamiltonian is given by $H_k = v_0(\mathbf{k} \times \sigma)$, with velocity $v_0 = 3.0$ eV chosen to be intermediate between $Bi_2Se_3$ and $Bi_2Te_3$. The real space basis includes two spin-degenerate states per lattice site. The kinetic Hamiltonian is diagonalized on this basis, and the resulting momentum-space basis is reduced by applying a high energy cutoff $W = 0.4$ eV to exclude high energy states that would not conform to a Dirac dispersion.

Defects are created by perturbing sites at selected coordinates $i$ with a scalar energy term. as $H_d = \sum_i U n_i$, where $n_i$ is the number operator, and the full Hamiltonian $H = H_k + H_d$ is diagonalized. The convention of creating point defects as triangular three-site clusters with a perturbation strength of $U/3$ on each site is adopted to give three-fold rotational symmetry, as seen experimentally[14,15], though calculated spectra do not strongly differentiate between differently sized small site clusters (Supplementary Fig. 6). The Hamiltonian is numerically diagonalized, and the ARPES spectral function is defined as the energy- and momentum-resolved spectral function of single particle annihilation, convoluted by a 10 meV half-width Lorentzian:

$$I(E, \mathbf{k}) = \sum_{f,\alpha,\sigma} |\langle f | a_{\mathbf{k},\sigma} | \alpha \rangle|^2 \times \frac{5 \text{ meV}/\pi}{(E - E(\alpha))^2 + (5 \text{ meV})^2} \quad (1)$$

This represents the spectral function for photoemission with the $z$-axis photon polarization, which acts with even reflection symmetry and removes

electrons from the dominant $p_z$ orbital component of the $Bi_2Se_{3-x}Te_x$ surface state. Energy-momentum dispersions were found to be rotationally isotropic, and have been rotationally averaged to obtain a small spacing between states on the momentum axis. Figures in the paper use the rotationally averaged 'radial' momentum ($K_r$), and single-axis spectral functions can be found in Supplementary Fig. 5. For smaller system sizes considered in Fig. 4, it was necessary to average over several randomly generated defect configurations to achieve convergence of the twist velocity and participation ratio. Additional modelling details are found in Supplementary Note 3.

**Metrics of localization.** Adding a phase twist $\theta$ at the system boundary provides an easy way to identify the degree to which electronic states are localized. This is achieved by the standard method of shifting the momentum (K-space) basis of the Hilbert space by an offset of $\mathbf{K}_\theta = \theta_x/L_x \hat{\mathbf{k}}_\mathbf{x} + \theta_y/L_y \hat{\mathbf{k}}_\mathbf{y}$, where $L_x$ ($L_y$) is the width of the system along the $x- $ ($y-$) axis. A velocity termed the 'twist velocity' can be calculated for individual eigenstates $|\alpha\rangle$ as $\mathbf{v}_\theta(\alpha) = \nabla_{\mathbf{K}_\theta} E_\alpha$ (using numerical derivatives), and represents the velocity of a persistent current. The energy- and momentum-resolved twist velocity ($v_\theta(E, \mathbf{k})$) displayed on ARPES spectral intensity maxima in Fig. 3 is calculated via a weighted average of states within $\pm 10$ meV of the energy E, as:

$$\mathbf{v}_\theta(E, \mathbf{k}) = \frac{\sum_{\alpha,\sigma} \mathbf{v}_\theta(\alpha) |\langle \mathbf{k}, \sigma | \alpha \rangle|^2}{\sum_{\alpha,\sigma} |\langle \mathbf{k}, \sigma | \alpha \rangle|^2} \quad (2)$$

Here, $\langle \mathbf{k}, \sigma |$ represents the free particle eigenstates that have momentum $\mathbf{k}$. When divided by the width of the system L along the indicated axis, the phase twist has units of wave number. The numerical derivative is performed at a small but nonzero twist value of $\theta = \frac{\pi}{10}$ along the $\hat{\mathbf{k}}$ direction to break degeneracies and avoid possible finite size nesting effects.

The participation ratio is defined as:

$$P_\alpha = \sum_i n_{i,\alpha}^2, \quad (3)$$

where the sum is over all sites in the system, and $n_{i,\alpha}$ is the LDOS on site $i$ of $|\alpha\rangle$, which is an eigenstate of the full Hamiltonian.

**Data availability.** The data and source code that support the findings of this study are available from the corresponding authors on reasonable request.

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

## Acknowledgements

We are grateful for discussions with W. Wu, P. Hohenberg and Y.-D. Chuang. Work at Stanford University was supported by the Department of Energy Grant DE-AC02-76SF00515. R.R.B. was supported by Purdue University startup funds.

## Authors contributions

Y.X. and J.C. carried out the numerical modelling with assistance from H.H. and guidance from R.R.B. and L.A.W.; Y.X., J.C., L.M., R.R.B. and L.A.W. participated in the analysis, figure planning and draft preparation; Z.A. and A.K. performed and analysed the STM experiments; L.A.W. was responsible for the conception and the overall direction, planning and integration among different research units.

## Additional information

**Competing financial interests:** The authors declare no competing financial interests.

**Publisher's note**: 

