## [Peer Review File · Nature Communications]

Reviewers' Comments:

Reviewer #1 (Remarks to the Author):

The manuscript by Xu et al. reports a theoretical (numerical simulations) study of defects in 3D topological insulator (TI) on the influence of their Dirac surface states observed in STM, ARPES and transport measurements. With a simple and convincing picture, the authors explain many results in past STM and ARPES studies on 3D TIs, e.g. the resonance defect states and the gap-like features at Dirac surface states, which provides a comprehensive view of the nature of topological surface states of 3D TIs. I think this work is of great help to the researchers in this field, especially for understanding the transport properties of 3D TIs. I recommend its publication in Nature Communications.

Nevertheless there is one point that should be clarified. In page 6, the last paragraph, the authors mentioned that "...the twist velocities of band features at energies $E > 0$ eV have similar amplitudes, and are approximately proportional to the apparent slope in momentum space, suggesting that all of these features are viable bands for charge transport. ". It is true for Fig. 3a. But for Figs. 3b and 3c, the velocities around E^* are clearly reduced. I imagine it will greatly lower the conductivity when Fermi level is around here. The authors should discuss the transport properties in this case.

Reviewer #2 (Remarks to the Author):

This review is for the article "Disorder enabled band structure engineering of a topological insulator surface." In this work, which appears primarily modeling/theoretical in nature, the authors claim that defect resonance states can give rise to an emergent surface bandstructure on the surface of single-Dirac cone topological insulators.

The authors use a simplified model of a Dirac cone surface state to model the effects of time-reversal invariant disorder. The paper claims that resonant defect states can coherently give rise to transport-active "impurity bands." I find the concept to be interesting, but not of broad interest. The role of the experiment is also unclear. From what I could tell the only experimental data shown was an STM image showing a defect resonance, and two curves in Figure 2 which serve to provide corroborating evidence that the simple surface model is providing reasonable results. From the ability to match these two experimental curves the authors claim that the model can provide quantitative results.

I have some specific criticisms of the work.

1. The model only includes the surface Hamiltonian and no bulk physics. In some of the dichalcogenide TIs the bulk Dirac point is below or the valence band. Furthermore, in cases where the Dirac point is accessible the strong defect potential chosen for fitting in the article may cause the bulk to react in some way which is not captured by the chosen model. Maybe these effects aren't important, but since the bulk is usually doped by vacancies in these materials and tends to dominate transport, it isn't clear.

2. Showing that the modeled resonant states have some measures of delocalization is important for the conclusions, but there is no clear evidence that they are important for transport in realistic systems. One naive point is that the localization length could be larger than the tested system sizes, but it could also be that transport is dominated by other channels, e.g. bulk conduction, other surface channels etc.

3. From the ability to fit the two experimental LDOS curves it is not clear how the authors can claim the model and results should quantitatively apply to considerations of transport.

4. There are arguments for bandstructure engineering. As written it is not clear how the authors plan to precisely control this from a materials standpoint. I did not even find mention of what materials process give rise to the defects which they consider.

While I think that the overall idea is interesting for specialists, I do not find the results, even if all are true, to be of enough interest to a broader audience to be published in Nature Communications. I think that the work is much more suitable for a specialized journal. Furthermore I believe that the conclusions are too strong for the particular calculations that have been carried out. The evidence here makes it plausible that these states could contribute to transport physics in real materials, but I do not find it completely convincing as currently written. Furthermore, if technological applications are to be the selling point I find there to be an inadequate discussion of the processes involved to use this effect.

Reviewer #3 (Remarks to the Author):

The manuscript by Y.Xu et al. describes forming of a new impurity band from the Dirac surface states of topological insulators. Usual impurity bands, for example in magnetic semiconductors like GaMnAs, should have localized carriers while novelty of the impurity bands based on the surface state of TIs would be lack of localization, and therefore conducting properties.

The authors use the surface state Hamiltonian and perturbing defects described by Hubbard-like Hamiltonian to describe the system. They predict that the resonant impurities could change band structure of the Dirac surface, depending on the concentration and strength of the impurity potential. In particular, the strong disorder potential could generate impurity band with the upper band separated by a gap from the lower band. This could explain opening of the gap in the Dirac spectrum observed often in ARPES, even for the scalar impurities.

I think that the paper is very well written, and gives a new interpretation to the problem with the ARPES data with strong scalar disorder, and therefore could be of interest for a broad audience reading Nature Communications.

I have, however several points, which should be clarified/extended in the revised version:

- 1) Although in the abstract it is written "Here we use numerical simulations and scanning tunneling microscopy data", as far as I can see there are only two experimental curves in Figs.2b and 2c. If the paper should be a comparison of the experimental and data with the simulations, it would be nice to try to increase the number of experimental curves
- 2) The question connected with the point (1), the authors find that the impurities with $U=-45\text{eV}$ fit this particular experimental curve. Why should one focus on this value? I would expect distribution of the different strengths of the potential depending on the sample.
- 3) In the context of point (2), is this possible to engineer the disorder? Is not it that depending on the sample, the impurity band will appear or disappear depending how strong defect potentials appear in the sample? How universal and tunable is this behavior?
- 4) Finally, why does the lower band of the Dirac structure survive, and only the upper branch is destroyed? This appears in simulations in Figs 2d-2e.
- 5) Some more details about calculation skim would be useful in the Methods or Supplementary. Do the authors average other impurity configurations as this is usually applicable?

Reviewer #1 (Remarks to the Author):

The manuscript by Xu et al. reports a theoretical (numerical simulations) study of defects in 3D topological insulator (TI) on the influence of their Dirac surface states observed in STM, ARPES and transport measurements. With a simple and convincing picture, the authors explain many results in past STM and ARPES studies on 3D TIs, e.g. the resonance defect states and the gap-like features at Dirac surface states, which provides a comprehensive view of the nature of topological surface states of 3D TIs. I think this work is of great help to the researchers in this field, especially for understanding the transport properties of 3D TIs. I recommend its publication in Nature Communications.

We are grateful for this summary, and for the recommendation.

Nevertheless there is one point that should be clarified. In page 6, the last paragraph, the authors mentioned that "...the twist velocities of band features at energies $E > 0$ eV have similar amplitudes, and are approximately proportional to the apparent slope in momentum space, suggesting that all of these features are viable bands for charge transport. ". It is true for Fig. 3a. But for Figs. 3b and 3c, the velocities around E^* are clearly reduced. I imagine it will greatly lower the conductivity when Fermi level is around here. The authors should discuss the transport properties in this case.

This point is now directly addressed on page 7 of the main text:

"However, twist velocities decay towards zero as the upper D_0 Dirac cone disperses into the

seemingly gapped region beneath the D_1 feature, particularly at momenta of $k \sim 0.05 \text{ \AA}^{-1}$, and this may greatly limit carrier mobility for certain values of the chemical potential (see comparison of twist and band velocities in the SI [25]).”

A figure has also been added to the Supplementary Information, tracing the twist velocity and impurity band slope in Fig. 3c as a function of momentum.

Reviewer #2 (Remarks to the Author):

This review is for the article "Disorder enabled band structure engineering of a topological insulator surface." In this work, which appears primarily modeling/theoretical in nature, the authors claim that defect resonance states can give rise to an emergent surface bandstructure on the surface of single-Dirac cone topological insulators.

The authors use a simplified model of a Dirac cone surface state to model the effects of time-reversal invariant disorder. The paper claims that resonant defect states can coherently give rise to transport-active "impurity bands." I find the concept to be interesting, but not of broad interest. The role of the experiment is also unclear. From what I could tell the only experimental data shown was an STM image showing a defect resonance, and two curves in Figure 2 which serve to provide corroborating evidence that the simple surface model is providing reasonable results. From the ability to match these two experimental curves the authors claim that the model can provide quantitative results.

I have some specific criticisms of the work.

1. The model only includes the surface Hamiltonian and no bulk physics. In some of the dichalcogenide TIs the bulk Dirac point is below or the valence band. Furthermore, in cases where the Dirac point is accessible the strong defect potential chosen for fitting in the article may cause the bulk to react in some way which is not captured by the chosen model. Maybe these effects aren't important, but since the bulk is usually doped by vacancies in these materials and tends to dominate transport, it isn't clear.

Yes, we agree that these real-world elements of the surface physics are extremely important to consider in the development of devices that make use of the “Dirac transport regime” of Bi_2Se_3 -family TIs. We have added some commentary on this (see responses further below), and listed several papers that focus more specifically on synthesis and device design (see the bottom of page 4). Ref. [41] (Checkelsky et al., Nature Physics 8, 729 [2012]) in particular demonstrates the use of electrostatic gating to lower the chemical potential and induce a phase transition driven by energetics of the lower Dirac cone.

With respect to the accuracy of the model we have implemented, the bulk electronic structure does factor in as a source of renormalization to the ‘U’ parameter, and this is in part why it was necessary to fit the U parameter from experimental data. We have added a further comparison with energetics in an STM surface map that contains a large number of

defects (Fig. 3e-f) to confirm that the model is not just contextually correct for an isolated defect, but accurately reproduces the energy scale of coherent resonance state physics for a large collection of defects.

2. Showing that the modeled resonant states have some measures of delocalization is important for the conclusions, but there is no clear evidence that they are important for transport in realistic systems. One naive point is that the localization length could be larger than the tested system sizes, but it could also be that transport is dominated by other channels, e.g. bulk conduction, other surface channels etc.

The development of real devices that incorporate topological surface state transport properties is one of the ongoing ‘grand goals’ of the field, and there have been significant successes. We have added citations (Ref. [39-43]) that address issues important for device development. These address how the chemical potential can be tuned through bulk/surface doping or electrostatic gating to access the electronic states considered in this paper, and to minimize bulk conductivity.

The first section of the SI also now contains a discussion of the defect types and distributions found at the surface of crystals synthesized in different ways.

3. From the ability to fit the two experimental LDOS curves it is not clear how the authors can claim the model and results should quantitatively apply to considerations of transport.

We have added a real-space comparison with the energetics measured by STM of a large 60X60nm defect-rich surface (Fig. 3e-f). The good correspondence found in this comparison provides an important corroboration that electrons near the Dirac point (the electrons that are significant for the “Dirac transport regime”) occupy coherent states that are delocalized between defect resonances. The simulation very accurately captures the energy scale of real space DOS fluctuations seen by STM, and can be plotted nicely on the same energy-resolving color scale as the experimental data, with no adjustment to the previously determined modeling parameters.

4. There are arguments for bandstructure engineering. As written it is not clear how the authors plan to precisely control this from a materials standpoint. I did not even find mention of what materials process give rise to the defects which they consider.

This is now outlined on page 4, as follows:

“Intensive characterization of Bi_2Se_3 has found that the density and type-range of defects can be tuned over a broad range by adjusting sample growth conditions [39]. For Se-poor and stoichiometric synthesis, defects near a cleaved surface are predominantly selenium vacancies positioned in the center of the outermost quintuple layer of the lattice. Interstitial atoms and replacement defects can be common at the surface of Se-rich or quench-cooled samples. The chemical potential can be shifted as a separate parameter via methods that

include bulk doping, surface dosing, and electrostatic gating [40–43].”

We have also added a discussion to the first section of the SI (also noted above) of different defect species and how they can be achieved in synthesis.

While I think that the overall idea is interesting for specialists, I do not find the results, even if all are true, to be of enough interest to a broader audience to be published in Nature Communications. I think that the work is much more suitable for a specialized journal. Furthermore I believe that the conclusions are too strong for the particular calculations that have been carried out. The evidence here makes it plausible that these states could contribute to transport physics in real materials, but I do not find it completely convincing as currently written. Furthermore, if technological applications are to be the selling point I find there to be an inadequate discussion of the processes involved to use this effect.

We nonetheless appreciate that the Reviewer has taken the time to provide detailed comments, which have been useful in revising the manuscript. We wish to stress that the main conclusion of this manuscript is that defect resonance states at a TI surface *enable a fundamentally new approach to band structure engineering*. Applied research is of tremendous importance; however, it is our perspective that in this case, the need to establish the novel underlying physical regime with respect to theoretical metrics and STM data precludes a serious focus on specific applied scenarios and “technological applications”.

Reviewer #3 (Remarks to the Author):

The manuscript by Y.Xu et al. describes forming of a new impurity band from the Dirac surface states of topological insulators. Usual impurity bands, for example in magnetic semiconductors like GaMnAs, should have localized carriers while novelty of the impurity bands based on the surface state of TIs would be lack of localization, and therefore conducting properties. The authors use the surface state Hamiltonian and perturbing defects described by Hubbard-like Hamiltonian to describe the system. They predict that the resonant impurities could change band structure of the Dirac surface, depending on the concentration and strength of the impurity potential. In particular, the strong disorder potential could generate impurity band with the upper band separated by a gap from the lower band. This could explain opening of the gap in the Dirac spectrum observed often in ARPES, even for the scalar impurities. I think that the paper is very well written, and gives a new interpretation to the problem with the ARPES data with strong scalar disorder, and therefore could be of interest for a broad audience reading Nature Communications.

I have, however several points, which should be clarified/extended in the revised version:
1) Although in the abstract it is written “Here we use numerical simulations and scanning tunneling microscopy data”, as far as I can see there are only two experimental curves in Figs.2b and 2c. If the paper should be a comparison of the experimental and data with the

simulations, it would be nice to try to increase the number of experimental curves

We have added a real-space comparison with the energetics measured by STM of a large 60X60nm defect-rich surface (Fig. 3e-f). The good correspondence found in this comparison provides an important corroboration that electrons near the Dirac point occupy coherent states that are delocalized between defect resonances. The simulation very accurately captures the energy scale of real space DOS fluctuations seen by STM, and can be plotted nicely on the same energy-resolving color scale as the experimental data, with no adjustment to the previously determined modeling parameters.

It may be worth noting that experimental data are also included in the inset of Fig. 1a, in which it is confirmed that the large ~8 nm diameter the defect resonance state is reproduced by the calculation.

2) The question connected with the point (1), the authors find that the impurities with $U=-45\text{eV}$ fit this particular experimental curve. Why should one focus on this value? I would expect distribution of the different strengths of the potential depending on the sample.

This is in principle a very valid point. However, it turns out that defects at the surface of Bi_2Se_3 are much more consistent in type than one would expect: they are predominantly Se vacancies found in the 2nd Se layer from the cleaved or grown crystal surface (middle of the 1st quintuple layer). This is discussed in Ref. [39], and the first paragraph of the Results section now notes that:

“Intensive characterization of Bi_2Se_3 has found that the density and type-range of defects can be tuned over a broad range by adjusting sample growth conditions [39]. For Se-poor and stoichiometric synthesis, defects near a cleaved surface are predominantly selenium vacancies positioned in the center of the outermost quintuple layer of the lattice. Interstitial atoms and replacement defects can be common at the surface of Se-rich or quench-cooled samples. The chemical potential can be shifted as a separate parameter via methods that include bulk doping, surface dosing, and electrostatic gating [40–43].”

We have therefore focused on just one U value, which appears to accurately address the Se-poor regime. For greater generality, we have now also added density-dependent twist velocity calculations for alternative U values ($U=-22.5$, $U=-90$) to the Supplemental Information (SI Fig. S2).

3) In the context of point (2), is this possible to engineer the disorder? Is not it that depending on the sample, the impurity band will appear or disappear depending how strong defect potentials appear in the sample? How universal and tunable is this behavior?

Yes, defect density can be tuned over a wide range without changing the defect type (see also the answer to the previous question, which includes the related textual revision). Further discussion of this has also been added to the first section of the SI. Extremely high Se vacancy

densities on the order of ~0.5% (3X what we have considered here) can be achieved in otherwise well ordered ~100nm patches of a sample surface, though larger scale crystallinity tends to suffer at such high densities.

4) Finally, why does the lower band of the Dirac structure survive, and only the upper branch is destroyed? This appears in simulations in Figs 2d-2e.

This is because the defects we have focused on are energetically located in the upper Dirac cone for a negative defect potential. The paper notes this with respect to the $E_R - E_D \sim U^{-1}$ resonance energy of isolated defects, and it fits in a consistent framework with the observation on page 7 that:

“Velocities are relatively large in the lower D_0 Dirac cone, where states are energetically distant from the resonance states, and maximal velocities are found immediately beneath D_0 , where elastic scattering is minimal due to the vanishing density of states near a 2D Dirac point.”

Certain other details of the dispersion, such as evidence of a faint ‘lower Dirac cone’ branch to the D_1 band, are now commented on in the SI.

5) Some more details about calculation skim would be useful in the Methods or Supplementary. Do the authors average other impurity configurations as this is usually applicable?

Yes, a few of the calculations average over configurations. This was not needed for the spectral function plots (E vs K) and other plots for large (~350X350) surfaces, but was essential to obtain smooth curves for the participation ratio and twist velocity at smaller system sizes (Fig. 4 panels). We have noted this in the methods section, and added further details about the calculation in the Supplementary Information.

Reviewers' Comments:

Reviewer #1 (Remarks to the Author):

The authors have answered all my questions satisfactorily. I now agree on publication of the work in Nature Communications.

Reviewer #2 (Remarks to the Author):

[The referee does not recommend the work for publication, with no further comments for the authors]

Reviewer #3 (Remarks to the Author):

[The reviewer recommends publication with no further comments for the authors]